# Personalized Medicine in Severe Asthma: From Biomarkers to Biologics

**DOI:** 10.3390/ijms25010182

**Published:** 2023-12-22

**Authors:** Chun-Yu Chen, Kang-Hsi Wu, Bei-Cyuan Guo, Wen-Ya Lin, Yu-Jun Chang, Chih-Wei Wei, Mao-Jen Lin, Han-Ping Wu

**Affiliations:** 1Department of Emergency Medicine, Tungs’ Taichung Metro Harbor Hospital, Taichung 435403, Taiwan; yoyo116984@gmail.com (C.-Y.C.); ambrose.alma@yahoo.com.tw (C.-W.W.); 2Department of Nursing, Jen-Teh Junior College of Medicine, Nursing and Management, Miaoli 35664, Taiwan; 3Department of Pediatrics, Chung Shan Medical University Hospital, Taichung 40201, Taiwan; cshy1903@gmail.com; 4School of Medicine, Chung Shan Medical University, Taichung 40201, Taiwan; 5Department of Pediatrics, National Cheng Kung University Hospital, College of Medicine, National Cheng Kung University, Tainan 70403, Taiwan; gbc628@gmail.com; 6Division of Pediatric Emergency Medicine, Department of Pediatrics, Taichung Veteran General Hospital, Taichung 43503, Taiwan; wylin002@gmail.com; 7Laboratory of Epidemiology and Biostastics, Changhua Christian Hospital, Changhua 500, Taiwan; 83686@cch.org.tw; 8Division of Cardiology, Department of Medicine, Taichung Tzu Chi Hospital, The Buddhist Tzu Chi Medical Foundation, Taichung 42743, Taiwan; 9Department of Medicine, College of Medicine, Tzu Chi University, Hualien 97002, Taiwan; 10College of Medicine, Chang Gung University, Taoyuan 333, Taiwan; 11Department of Pediatrics, Chiayi Chang Gung Memorial Hospital, Chiayi 61363, Taiwan

**Keywords:** severe asthma, biomarkers, biologics, personalized medicine

## Abstract

Severe asthma is a complex and heterogeneous clinical condition presented as chronic inflammation of the airways. Conventional treatments are mainly focused on symptom control; however, there has been a shift towards personalized medicine. Identification of different phenotypes driven by complex pathobiological mechanisms (endotypes), especially those driven by type-2 (T2) inflammation, has led to improved treatment outcomes. Combining biomarkers with T2-targeting monoclonal antibodies is crucial for developing personalized treatment strategies. Several biological agents, including anti-immunoglobulin E, anti-interleukin-5, and anti-thymic stromal lymphopoietin/interleukin-4, have been approved for the treatment of severe asthma. These biological therapies have demonstrated efficacy in reducing asthma exacerbations, lowering eosinophil count, improving lung function, diminishing oral corticosteroid use, and improving the quality of life in selected patients. Severe asthma management is undergoing a profound transformation with the introduction of ongoing and future biological therapies. The availability of novel treatment options has facilitated the adoption of phenotype/endotype-specific approaches and disappearance of generic interventions. The transition towards precision medicine plays a crucial role in meticulously addressing the individual traits of asthma pathobiology. An era of tailored strategies has emerged, allowing for the successful targeting of immune-inflammatory responses that underlie uncontrolled T2-high asthma. These personalized approaches hold great promise for improving the overall efficacy and outcomes in the management of severe asthma. This article comprehensively reviews currently available biological agents and biomarkers for treating severe asthma. With the expanding repertoire of therapeutic options, it is becoming increasingly crucial to comprehend the influencing factors, understand the pathogenesis, and track treatment progress in severe asthma.

## 1. Introduction

Asthma, one of the most common chronic inflammatory respiratory diseases, affects more than 300 million adults and children worldwide [1,2]. Asthma was previously considered a single diagnosis with standardized treatment for all patients; however, it is currently recognized as a heterogeneous, multifactorial disease involving multiple genetic and environmental factors [3]. The four key factors that play important roles in its pathogenesis are T-helper type-2 (Th2)-high inflammation, Th2-low inflammation, airway hyperresponsiveness (AHR), and airway remodeling [4]. Asthma is a general diagnostic term that describes various clinical presentations (phenotypes) and diseases with distinct mechanistic pathways (endotypes). Common asthma phenotypes included in the Global Initiative for Asthma (GINA) report include allergic asthma, non-allergic asthma, late-onset asthma, asthma with persistent airflow limitation, and asthma with obesity [5]. Asthma phenotyping is becoming increasingly important because of the development of new therapies targeting specific inflammatory mediators. In recent years, asthma phenotypes have been classified into two asthma endotypes: type-2 (T2)-high and T2-low [6]. The T2-high endotype is characterized by Th2-associated cytokines such as interleukin (IL)-4, IL-5, and IL-13 and elevated levels of eosinophils, serum periostin, and total immunoglobulin E (IgE) and includes allergic and eosinophilic asthma [7,8]. The T2-low endotype has no identified biomarkers or eosinophilic elevation and generally includes all patients with asthma without Th2-high inflammation [8,9].

Clinically, asthma is characterized by variable signs and symptoms, including shortness of breath, chest tightness, coughing, recurrent wheezing, and airflow limitation. Although the majority of patients with asthma can achieve good disease control using standard treatments, approximately 5–10% of the global asthmatic population experiences various subtypes of inadequately controlled and difficult-to-treat asthma [10].

The European Respiratory Society (ERS) and the American Thoracic Society (ATS) jointly define severe asthma as a condition that can only be controlled by high dosages of inhaled corticosteroid (ICS)/long-acting β2-adrenergic agonist (LABA) combinations, along with the addition of other drugs (i.e., tiotropium, leukotriene modifiers, oral corticosteroids (OCSs)); in worse cases, asthma might not be adequately controlled despite such aggressive treatments [11]. In addition, patients with severe asthma may consume a large proportion of overall asthma resources, and preventing exacerbations is an important unmet issue. Over the past two decades, an increasing understanding of the complex pathophysiology of asthma and its associated inflammatory mechanisms has led to the development of personalized biologics for treating patients with severe asthma.

This review focuses on the role of molecular biomarkers in severe asthma and recent clinical advances for severe asthma. Additionally, the importance of classifying patients according to different endotypes as well as new biological therapies for severe asthma are discussed.

## 2. Pathogenesis of Severe Asthma

The pathogenesis of asthma is a complex process in which genetic and environmental factors play important roles, leading to a reduction in airway diameter. According to previous studies, there are four important components in the pathogenesis of asthma: Th2-high inflammation, Th2-low inflammation, AHR, and airway remodeling [4]. Asthma clinically manifests as distinct phenotypes driven by complex pathobiological mechanisms known as endotypes. Most patients with asthma have T2 inflammation, characterized by cytokines (IL-4, IL-5, and IL-13) and inflammatory cells (eosinophils, mast cells, basophils, and IgE-producing plasma cells) (Table 1) [12]. In particular, T2-high eosinophilic inflammation is common in patients with allergic or non-allergic asthma and is associated with a high risk of severe asthma and frequent asthma exacerbations [13,14]. Currently, a few clinical biomarkers are available such as IgE, blood or sputum eosinophil count, and fractional exhaled nitric oxide (*F*_eNO_) [15]. Combining biomarkers with T2-targeting monoclonal antibodies (mAbs) is critical for developing personalized treatment strategies for severe asthma [16,17].

### 2.1. T2-High Asthma

Allergic asthma is triggered by allergens, pollutants, and microorganisms, which are captured by dendritic cells (DCs) and drive bronchial epithelial cells (BECs) to release IL-25, IL-33, and thymic stromal lymphopoietin (TSLP), which in turn activate group 2 innate lymphoid cells (ILC2) in the bronchial mucosa [18]. Bronchial epithelial IL-25 expression plays an important role in triggering type-2 responses; patients with higher IL-25 levels show more severe AHR, increased sputum and blood eosinophils, higher serum IgE levels, thickening of the subepithelial cell layers to a greater extent, and higher expression of Th2 signature genes [19]. IL-33 induces the expression of fucosyltransferase 2 (Fut2), and glycation of BECs leads to sustained ILC2 activation in T2 asthma [18]. Cytokines produced by BECs promote functioning of DCs, polarize CD4+ T cells, and promote polarization of Th2 cells [20]. ILC2 and Th2 cells share many features, such as the expression of cytokine receptors, transcription factor GATA3, and CRTH2 (chemokine receptors expressed by Th2 cells), and produce IL-4, IL-5, IL-9, and IL-13 [21,22]. The functions of ILC2 and Th2 cells overlap and are thought to play important roles in the pathogenesis of asthma.

BECs also release progranulin, which in turn activates fibroblast and natural killer T (NKT) cells to produce IL-4 [18]. Activated ILC2 may also produce IL-4 under certain conditions or in certain tissues; however, the details remain unknown.

T follicular helper (T_FH_) cells control IgE synthesis by secreting IL-4 to allergen-specific B cells [20,23]. Eosinophils and CD4+ cells producing IL-5 are frequently found in the blood and lung lavage fluid of patients with asthma [24]. IL-5 drives the development and activation of airway eosinophils and promotes eosinophilic inflammation. Moreover, IL-5 stimulates BECs and contributes to the loss of epithelial integrity [20,23]. IL-9 is a key factor in allergic airway inflammation and is mainly derived from Th9, ILC2, and Th2 cells [25]. IL-9 promotes the pathogenesis of asthma by activating and recruiting mast cells and eosinophils, enhancing B-cell IgE production, promoting goblet cell metaplasia, increasing epithelial cell mucus production, and triggering AHR [20,23,25]. On the other hand, IL-13 is mainly involved in mucus production, airway remodeling, and bronchial hyperresponsiveness [20,23,26]. Furthermore, in patients with T2-high asthma, such a complex process may result in IgE, IL-13, IL-4, IL-5, and eosinophil responses, covering more than 50% of asthma endotypes [15].

### 2.2. T2-Low Asthma

Type-2-low asthma is characterized by a lack of type-2 biomarkers, later age at onset of the disease, presence of neutrophils, obesity, and/or unresponsiveness to corticosteroids [27]. The key cytokines involved in T2-low neutrophilic asthma include IL-17, IL-8, IL-6, and IL-1β.

The main features of Th1 inflammation are increased Th1 cells and infiltration of characteristic cytokines, including interferon (IFN)-γ and tumor necrosis factor (TNF)-α, which have a significant effect on airway smooth muscle (ASM) [28]. Th17 cells produce IL-17A, IL-17F, IL-21, IL-22, and TNF-α to perform their functions, and IL-17A and IL-17F are linked to severe asthma with neutrophil inflammation and responsible for corticosteroid resistance in asthma [20]. Furthermore, IL-17A and IL-17F induce BECs and subepithelial airway fibroblasts to release potent neutrophil chemoattractants, including IL-8 (CXCL8) and CXCL1/GRO-α [29,30,31]. In addition, large amounts of ILC3 have been found in the bronchoalveolar lavage fluid (BALF) of adults with severe asthma and in the blood of obese children with asthma [31,32].

### 2.3. Phenotype Overlap

Throughout the lifespan of individuals with asthma who develop a T2 immune and inflammatory response, a variety of environmental stimuli can bring about alterations in the pre-existing inflammatory profile. This can result in the convergence of mixed molecular pathways and the overlap of phenotypes. Several mediators regulate both eosinophil and neutrophil recruitment to the airways. In addition, mixed inflammatory phenotypes of eosinophils and neutrophils are known to play an important role in severe asthma. Numerous double-positive Th2/Th17 lymphocytes secreting large amounts of IL-4 and IL-17 have been detected in the BALF of patients with severe asthma [33]. A previous study has reported that Th2 and Th17 lymphocytes are involved in the induction of severe experimental asthma in mice [34]. However, cellular pathophysiology and phenotype require further studies to characterize specific asthma phenotypes in severe asthma.

## 3. Traditional Asthma Biomarkers in Clinical Practice

Currently, a few clinical biomarkers for asthma are available, including eosinophils, neutrophils, IgE, FeNO, leukotrienes, and periostin. Measurable indicators may link an endotype with a phenotype, and many other asthma biomarkers are under investigation, such as cytokines, dipeptidyl peptidase 4, and volatile organic compounds [3]. Although biomarker-directed management of asthma remains limited, the application of these traditional and novel biomarkers in clinical practice may be necessary to further improve the response to specific biologics.

### 3.1. Eosinophils

Healthcare providers can measure eosinophil levels in the blood and sputum to assess disease severity, eosinophilic asthma diagnosis, and response to specific treatments. High eosinophil levels can be observed in both allergic and non-allergic asthma, and sputum eosinophilia is more accurate than serum eosinophilia [35]. Sputum eosinophil count can be used to guide therapy in patients with asthma, and serum eosinophil count can be used to monitor biochemical responses in patients treated with anti-IL-5 [36]. Furthermore, peripheral blood eosinophilia is associated with more frequent asthma exacerbations but cannot be used to monitor the response to anti-IL-4R therapy [37]. Nonetheless, serum eosinophil count remains one of the most useful biomarkers for asthma classification and assessing response to treatment.

### 3.2. Neutrophils

Some clinical forms of asthma are characterized by a high number of airway neutrophils; higher expression of bronchial neutrophils is observed in patients with severe asthma [38]. The Th17 endotype predominates in most cases of T2-low asthma, and elevated serum and sputum neutrophil levels are common. However, some patients with neutrophilic asthma do not have elevated peripheral neutrophilia, and there is no strong correlation between sputum and blood neutrophil levels [3].

### 3.3. IgE

The role of IgE is complex and is linked with several immune and structural cells involved in allergic asthma. Th2 cells communicate with naïve IgM+ B cells to mediate class switch recombination to IgE. Switched B cells release allergen-specific IgE, which sensitizes resident mast cells and recruits basophils for activation in the airway mucosa [18]. Although IgE levels do not accurately predict asthma treatment efficacy or disease severity, they may predict cases requiring retreatment after discontinuation [39,40]. Anti-IgE is indicated in patients with severe asthma who are sensitized to perennial allergens, particularly in childhood-onset asthma and in patients with coexisting conditions, such as allergic rhinitis, chronic rhinosinusitis with nasal polyposis, or chronic urticaria [41].

### 3.4. FeNO

Nitric oxide is mainly produced by airway epithelial cells expressing IL-4Rα and responding to IL-4 and IL-13 [3]. FeNO can be measured easily and non-invasively in the office and can be used to identify Th2-type asthma with eosinophilic airway inflammation [42]. However, FeNO levels should not be used as a universal guide for asthma therapy [36]. Blood eosinophil count and FeNO levels are the two most important type-2 biomarkers for the management of severe asthma. These biomarkers can be used to identify severe asthma and predict responses to targeted treatments.

### 3.5. Leukotrienes

Leukotrienes are generated via a pathway initiated by 5-lipoxygenase (5-LOX) enzyme. Cysteinyl leukotrienes are potent mediators of bronchoconstriction and pro-inflammatory condition [43]. Leukotrienes have been demonstrated to contract ASM, increase vascular permeability and mucus secretion, and attract and activate other inflammatory cells in the airways of patients with asthma. High urinary leukotriene E4 (LTE4) is associated with the disease severity and is a marker of T2 airway inflammation [44]. However, LTE4 has several limitations in clinical practice, including the logistical difficulty of collecting 24 h specimens, lack of extensive testing, and administration of leukotriene receptor antagonists, which may affect the results.

### 3.6. Periostin

Serum periostin is primarily produced by structural cells in the airways, such as fibroblasts, eosinophils, and BECs and is believed to play an important role in airway remodeling and inflammation. Periostin is relatively stable with little variation, and a strong correlation between FeNO and serum periostin levels has been observed in severe asthma [45]. Additionally, serum periostin is a useful biomarker for the management of severe asthma and can serve as a long-term predictive marker [45].

## 4. Biologic Therapies for Severe Asthma

Symptom-control-based treatment options for severe asthma in the past have been replaced by a personalized approach to medicine in the present. Identification of relevant inflammatory mechanisms enables clinicians to provide targeted and personalized biological treatments for patients with severe asthma. Currently, six targeted treatments for severe asthma have been approved by the US Food and Drug Administration (FDA), including omalizumab, mepolizumab, benralizumab, reslizumab, dupilumab, and tezepelumab (Table 2, Figure 1) [46].

### 4.1. Omalizumab

Omalizumab was the first biologic drug approved by the FDA for the treatment of severe asthma and was the longest and most widely used drug in adults and adolescents and, subsequently, in children as young as 6 years. It is an mAb with a dual mechanism of action; it not only prevents IgE from binding to its high-affinity receptor (FcεRI) but also inhibits mast cell IgE receptor expression [47]. Since 2004, omalizumab has been the first mAb to be included in Step 5 of the GINA recommendations as an addition to standard therapy and has been shown to be effective in improving asthma control, relieving symptoms, reducing exacerbation risk, improving lung function, and reducing healthcare costs [48,49,50]. A review of 25 clinical trials reported in 2014 showed that the exacerbation rates between weeks 16 and 60 were reduced by 26% in the placebo group and 16% in the omalizumab group [51]. The risk of hospitalization decreased from 3% in the placebo group to 0.5% in the omalizumab group between weeks 28 and 60 [51]. Patients receiving omalizumab were also more likely to reduce or withdraw their ICS dose than those receiving a placebo; however, there was no significant difference in OCS withdrawal [51]. Dosage of omalizumab is calculated based on the age and baseline free total IgE level, and it is suitable for children aged 6–11 years with total IgE levels between 30 and 1300 kU/L and patients aged 12 years and older with total IgE levels between 30 and 700 kU/L.

However, a recent study evaluating the progression of asthma symptoms and biomarkers in adults treated with omalizumab showed similar benefits in patients with T2-high and T2-low asthma and did not find any association between exacerbation rates and increased FeNO ≥ 25 ppb or eosinophil count ≥300 cells/μL [52]. It appears that omalizumab plays an important role in non-specific severe asthma, and IgE levels do not accurately predict the treatment response. Finally, omalizumab remains a good option for clinical treatment of severe asthma and certain allergic diseases, with a long-term efficacy, safety, and tolerability profile.

### 4.2. Mepolizumab

Mepolizumab, an IgG1 antibody directed against IL-5, was approved in 2015 as the first anti-IL-5 biologic for add-on maintenance therapy in patients with severe asthma. Mepolizumab has been demonstrated to significantly reduce asthma exacerbations by 53% in adults and adolescents with severe asthma and evidence of an eosinophilic phenotype with increased blood eosinophil counts ≥150 cells/μL [53]. Based on previous clinical trials (DREAM, SIRIUS, MUSCA, and MENSA), mepolizumab was found to reduce asthma exacerbation rates and OCS use, improve quality of life and symptom control, and slightly increase forced expiratory volume in 1 s (FEV1) [53,54,55,56]. Mepolizumab is administered as a subcutaneous injection at a dose of 100 mg every 4 weeks in children aged 6 years and above and is indicated for the treatment of severe asthma in patients with peripheral eosinophil count ≥150 cells/μL. In addition, mepolizumab is effective in both non-allergic and allergic patients with severe eosinophilic asthma, and switching from omalizumab to mepolizumab is recommended for patients not adequately controlling disease with anti-IgE therapy [57,58,59]. Finally, mepolizumab has a favorable safety profile, and patients with severe asthma and higher blood eosinophil count show a greater reduction in exacerbation rate.

### 4.3. Benralizumab

Benralizumab, registered in 2017, is an mAb targeting the alpha subunit of IL-5R rather than the entire cytokine, preventing IL-5 from exerting its effect on target cells (eosinophils, basophils, and ILC2) and triggering eosinophil apoptosis via antibody-dependent cell-mediated cytotoxicity [60,61]. Benralizumab is an FDA-approved biologic as add-on maintenance therapy for the treatment of severe asthma in patients aged 12 years and above with peripheral eosinophil count ≥150 cells/μL. Benralizumab is administered subcutaneously at a dose of 30 mg every 4 weeks for 3 doses, followed by a maintenance dose of 30 mg every 8 weeks. According to the ZONDA, SIROCCO, ANDHI, and CALIMA trials, benralizumab significantly decreased the exacerbation rate and OCS use as well as improved asthma symptoms, quality of life, and lung function [62,63,64,65,66]. Benralizumab was more effective in patients with eosinophilic uncontrolled asthma than in those with atopic asthma.

### 4.4. Reslizumab

Reslizumab, which targets IL-5, was approved in 2016 for use in patients aged 18 years and above with severe eosinophilic asthma with peripheral eosinophil count ≥400 cells/μL. Reslizumab is administered intravenously and is currently the only weight-based intravenous dosing regimen administered at a dose of 3 mg/kg every 4 weeks. Reslizumab improves FEV1, reduces asthma exacerbation, reduces absolute eosinophil count in the sputum and blood, and improves the quality of life in patients with uncontrolled eosinophilic asthma [67,68,69]. A previous study has shown greater effectiveness of reslizumab than that of benralizumab in selected patients [70]. Reslizumab has a good safety profile and is well tolerated, with only rare cases of anaphylactic reaction following injection.

### 4.5. Dupilumab

Dupilumab, the first fully human anti-IL-4 receptor mAb, approved in 2018 for add-on maintenance treatment of severe asthma, blocks both IL-4 and IL-13 from binding to IL-4Rα. It is currently approved for use in patients aged 6 years and above with severe asthma of eosinophilic phenotype or who are dependent on OCSs [71]. Several clinical trials have reported significant improvements in asthma control and lung function and reduced rates of asthma exacerbation and OCS use after treatment with dupilumab [72,73,74,75]. Additionally, it has shown beneficial effects in patients with both allergic and non-allergic asthma, and the baseline FeNO level is used as a predictor of clinical response to dupilumab.

### 4.6. Tezepelumab

Tezepelumab, the newest drug approved in 2021, is an IgG2 mAb that blocks TSLP and inhibits its interaction with receptors. It targets alarmins, which are key epithelial inflammatory cytokines involved in the pathogenesis of asthma, and is a novel group of antibodies (TSLP, IL-25, and IL-33). Tezepelumab is approval for add-on maintenance therapy in adults and adolescent patients aged 12 years and above with severe asthma. It is administered by subcutaneous injection at a dose of 210 mg every 4 weeks. According to the PATHWAY, NAVIGATOR, SOURCE, and DESTINATION trials, the clinical efficacy of tezepelumab has been demonstrated in patients with severe asthma with decreased exacerbation rate, increased prebronchodilator FEV1, improved quality of life, reduced OCS dose, and favorable long-term safety [76,77,78,79]. Additionally, as eosinophil counts and FeNO levels increased, so did their effectiveness in reducing asthma exacerbation.

### 4.7. The potential Combination of Biologics in Cases of Phenotype Overlap

The combination of two biologics presents a therapeutic strategy that can concurrently target different inflammatory pathways. This approach has been employed in patients experiencing uncontrolled severe asthma, particularly those demonstrating evidence of both allergic and eosinophilic phenotypes, or individuals with severe asthma and T2 comorbidities [80]. The majority of presently available asthma biologics focus on targeting specific cytokines or cells within the T2 inflammatory pathway, with the notable exception of the recently approved tezepelumab. Limited data currently exist regarding the combined use of biologics for treating severe uncontrolled asthma with phenotype overlap. As of now, no established guidelines recommend dual biologic therapy for these conditions. Ongoing research and clinical experience will provide further insights into the specific roles and benefits of tezepelumab in optimizing treatment outcomes for patients with severe asthma and complex immunopathological profiles.

## 5. Potential New Biologics in the Future

A deeper understanding of relevant inflammatory mechanisms will enable clinicians to provide a targeted, personalized biological therapy for patients with severe asthma. Although progressive research on mAbs for the treatment of severe asthma has been carried out in recent years, a large proportion of patients do not respond to treatment. Innate cytokines, known as alarmins, specifically TSLP, IL-25, and IL-33, are promising candidates for the development of novel biological therapies against severe asthma. The exploration of molecular targets within the IL-1β, IL-23, and IL-17 axis is shedding light on potential avenues for addressing type-2-low neutrophilic asthma. This deeper understanding will pave the way for the development of innovative biotherapeutics having the potential to address unmet needs of patients with severe asthma refractory to conventional treatments.

### 5.1. Itepekimab and Astegolimab—IL-33

IL-33 activates fucosyltransferase 2, inducing fucosylated BECs that are critical for the sustained activation of ILC2, and cooperates with TSLP to promote type-2 immune/inflammatory responses [81]. Owing to its heterogeneity, IL-13 is a pleiotropic cytokine of cellular origin and downstream functions. In particular, IL-33 induces airway hyper-responsiveness by stimulating ILC2 and mast cells to release IL-13 [82]. A few clinical trials have shown promising results with anti-IL-33 drugs but studies with anti-IL-25 antibodies have not been conducted. There are a few ongoing phase 2 trials investigating biological drugs targeting IL-33 or its ST2 receptors in the context of severe asthma [83]. Of particular interest was the evaluation of itepekimab, an anti-IL-33 mAb, with a comparative analysis against the well-established dupilumab. Preliminary findings suggest that although itepekimab demonstrates efficacy in improving severe asthma symptoms, its therapeutic effects do not surpass those of dupilumab [83]. Notably, itepekimab exhibits a sustained reduction in blood eosinophil counts, indicating its potential as a viable treatment option for patients with moderate asthma [84]. Preliminary results from a phase 2 trial underscored the noteworthy impact of astegolimab on asthma exacerbation rate [85]. Subcutaneous administration of a dose of 70 mg emerged as a particularly effective regimen. Importantly, the trial cohort included patients with severe asthma, including those with low eosinophil counts, which are often challenging to manage using existing therapies [85]. This diversity of patient populations enhances the clinical relevance of astegolimab and indicates its potential as a versatile treatment option.

### 5.2. Fevipiprant–Prostaglandin D2 (PGD2)

PGD_2_, which primarily originates from mast cells, plays a pivotal role in the pathogenesis of type-2 asthma. Its pro-inflammatory actions are orchestrated through stimulation of the CRTH2 receptor expressed in Th2 lymphocytes, ILC2, and eosinophils [85]. The intricate interplay between PGD_2_ and CRTH2 serves as a focal point for understanding the mechanisms underlying type-2 asthma and presents a potential target for therapeutic intervention.

Fevipiprant, a notable player in the exploration of the PGD_2_ pathway, is a selective CRTH2 receptor antagonist. Unlike mAbs, fevipiprant is a small compound administered orally, positioning it as a unique therapeutic agent in the arsenal against type-2 asthma [86]. The selectivity of fevipiprant lies in its ability to block the binding of PGD_2_ to CRTH2, thereby disrupting the inflammatory cascade implicated in the pathophysiology [86]. Notably, fevipiprant induces a slight increase in FEV1, a functional effect comparable to that of the leukotriene receptor antagonist montelukast [87,88]. Despite these findings, the modest improvement in FEV1 prompted a closer examination of the potential of fevipiprant as a therapeutic agent and its distinctive role in the broader landscape of asthma management.

### 5.3. Other Targets

The forefront of severe asthma research is marked by ongoing early clinical trials investigating a spectrum of targets. Among these, the transcription factor GATA3 (SB010), prostaglandin D2 receptor (QAW039A), tyrosine kinase (imatinib), anti-IL-6 receptor (tocilizumab), and endothelin-A receptor (sitaxenten) have garnered attention for their potential impact on diverse molecular pathways associated with severe asthma [89,90,91,92,93]. This exploration reflects an evolving understanding of the complex nature of asthma pathogenesis. However, the challenge lies in reconciling the heterogeneity among patients, both phenotypically and endotypically, which may impact the effectiveness of targeted therapies.

## 6. Conclusions

Few mAbs have been evaluated for their efficacy, safety, and targeting pathways, such as anti-IgE, anti-IL-5, anti-IL-5 receptor, anti-TLSP, and anti-IL-4/IL-13 receptor. Enhanced comprehension of the heterogeneity, pathobiology, phenotypes, endotypes, and emerging therapeutic modalities in the context of asthma empowers physicians to deliver personalized medical interventions to individuals with severe asthma. Currently, the FDA has approved six biologics for the management of severe asthma. These biological agents have demonstrated substantial advancements in symptom control and improvement of lung function and overall health status of individuals afflicted with severe asthma of either the allergic or non-allergic eosinophilic T2-high type. Nevertheless, individuals with severe T2-low asthma do not generally experience therapeutic benefits as observed in patients with severe T2-high disease. Development of innovative biological treatments tailored specifically for severe neutrophilic or paucigranulocytic asthma remains a challenge. Moreover, investigation of the potential of combination therapies or the utilization of biomarkers to guide biological therapy introduces a novel and unexplored dimension, warranting focused attention and research to cater to the requirements of patients exhibiting insufficient responses to initial biologic treatments. Ultimately, the challenge lies in conducting studies that facilitate head-to-head comparisons, potentially contributing to enhanced biological efficacy in specific patient groups.

## Figures and Tables

**Figure 1 ijms-25-00182-f001:**
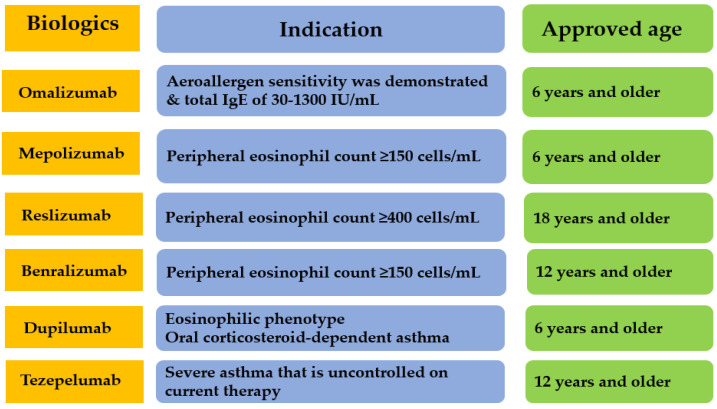
FDA-approved biological therapies for severe asthma based on indication and age.

**Table 1 ijms-25-00182-t001:** Summary of the current comprehension of type-2-high asthma and type-2-low asthma.

Characteristic	Type-2-High Asthma	Type-2-Low Asthma
**Phenotypes**	Early-onset allergic asthma Late-onset Complex T2-high asthma	Late-onset airway disease Early-onset Late-onset Obese Infection Smoking
**Disease severity**	Mild to very severe disease	Mild disease with intermittent to moderate obstruction
**Response to therapy**	Responsive to ICS Less responsive to ICS Refractory to ICS	Likely unresponsiveness to corticosteroids
**Mediators**	IL-4, IL-5, IL-13, IgE, INF-γ, eosinophils, mast cells, basophils	IL-1, IL-6β, IL-8, IL-17, neutrophils

**Table 2 ijms-25-00182-t002:** FDA-approved biological therapies for severe asthma.

Biologic Therapies	Omalizumab	Mepolizumab	Benralizumab	Reslizumab	Dupilumab	Tezepelumab
**Targets**	IgE	IL-5	IL-5Rα	IL-5	IL-4Rα	TSLP
**Related cell**	T_FH_ cell	Th2 cell	Th2 cell	Th2 cell	Th2 cell	Epithelial cell
**Molecular mechanisms**	Blocks IgE-mediated immune stimulation	Prevents binding of IL-5 to IL-5Rα	Blockade of IL-5Rα ADCC-induced eosinophil apoptosis	Prevents binding of IL-5 to IL-5Rα	Dual receptor antagonism of IL-4/IL-13	Prevents TSLP binding to its receptor complex
**Efficacy**	Exacerbation ↓ FEV1 ↑ Quality of life and symptom control ↑	Exacerbation ↓ FEV1 ↑ Blood and sputum eosinophils ↓ Quality of life and symptom control ↑ OCS intake ↓	Exacerbation ↓ FEV1 ↑ Blood and sputum eosinophils ↓ Quality of life and symptom control ↑ OCS intake ↓	Exacerbation ↓ FEV1 ↑ Blood eosinophils ↓ Quality of life and symptom control ↑ OCS intake ↓	Exacerbation ↓ FEV1 ↑ OCS intake ↓	Exacerbation ↓ Blood eosinophils ↓ FeNO ↓
**Approved** **ages**	≥6 years old	≥6 years old	≥12 years old	≥18 years old	≥6 years old	≥12 years old
**Route**	Subcutaneous	Subcutaneous	Subcutaneous	Intravenous	Subcutaneous	Subcutaneous

## Data Availability

No new data were created or analyzed in this study. Data sharing is not applicable to this article.

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
