# Peer review of "Personalized Medicine in Severe Asthma: From Biomarkers to Biologics"

_ijms, 2023, doi:10.3390/ijms25010182_

Round 1

Reviewer 1 Report

Comments and Suggestions for Authors

Thank you for the opportunity to review this really interesting and well written review with regards to personalized medicine in severe asthma. Authors in this article present in detail current approaches and future perspectives for therapeutic management with biologics of patients with severe asthma based on applicable biomarkers.

Authors should only comment in greater extent possible combination of biologics in cases of phenotype overlap.

Reviewer 2 Report

Comments and Suggestions for Authors

The submitted review manuscript addresses a very current topic, namely the diagnosis of severe asthma biomarkers and personalized therapy with biologicals. 

Many reviews have been written on this topic, so it is extremely important for the authors to show the uniqueness of their work. I strongly recommend writing Chapter 2 'Materials and Methods', where you write the criteria for inclusion and exclusion of articles in this review. In this chapter, you can clearly identify the type of review, for example, a literature review or content analysis, a critical review, or a narrative review.

I also recommend providing a graphical abstract of your review; this will help readers see all the different biologics that are or potentially are being used for severe asthma.
